# Reference curve sampling variability in one–sample log–rank tests

**Moritz Fabian Danzer**⊙°*, **Jannik Feld**°, **Andreas Faldum**, **Rene Schmidt**

Institute of Biostatistics and Clinical Research, University of Münster, Münster, Germany

☙ These authors contributed equally to this work.
* moritzfabian.danzer@ukmuenster.de

**Data Availability Statement:** All relevant data are within the paper and its Supporting information files.

**Funding:** The work of MFD was funded by the German Science Foundation (Deutsche

## Abstract

The one–sample log–rank test is the method of choice for single–arm Phase II trials with time–to–event endpoint. It allows to compare the survival of patients to a reference survival curve that typically represents the expected survival under standard of care. The one–sample log–rank test, however, assumes that the reference survival curve is known. This ignores that the reference curve is commonly estimated from historic data and thus prone to sampling error. Ignoring sampling variability of the reference curve results in type I error rate inflation. We study this inflation in type I error rate analytically and by simulation. Moreover we derive the actual distribution of the one–sample log–rank test statistic, when the sampling variability of the reference curve is taken into account. In particular, we provide a consistent estimate of the factor by which the true variance of the one-sample log–rank statistic is underestimated when reference curve sampling variability is ignored. Our results are further substantiated by a case study using a real world data example in which we demonstrate how to estimate the error rate inflation in the planning stage of a trial.

## Introduction

The one–sample log–rank test is the method of choice for single–arm Phase II trials with time–to–event endpoint. It allows to compare the survival of the patients to a prefixed reference survival curve that typically represents the expected survival under standard of care. First proposed by [1], its practical implementation including sample size calculation has been described by [2]. The one–sample log–rank test is often criticized in different directions. First, it has been reported repeatedly in the literature that the original one–sample log–rank test tends to be conservative (see [3, 4]). One reason for the test's inaccuracy is the dependence between the estimators of mean and variance of the original one–sample log–rank statistic when sample size is small. Several attempts have been made in the literature to correct for this (see [3–7]). Amongst those, the proposal made by [6] is presently implemented in the commercial software PASS [8] for sample size calculation for the one–sample log–rank test. Another more conceptual point of criticism against the one–sample log–rank test relates to the process of selecting the reference survival curve. It is common practice to choose the reference survival curve in the light of historic data on standard treatment. At the data level the difficulty

Forschungsgemeinschaft, DFG, https://www.dfg.
de, grant number 413730122). The funders had no
role in study design, data collection and analysis,
decision to publish, or preparation of the
manuscript.

**Competing interests:** The authors have declared
that no competing interests exist.

is that it might not reflect recent advances in diagnostics and/or concomitant therapy for standard of care thus resulting in a bias by not addressing confounders. Therefore, careful choice of the historic data set is crucial. At the level of analysis, the problem is that choosing the reference curve in the light of historic data implies that the reference survival curve itself is prone to sampling error. This sampling variability of the reference curve however, is ignored in the original one–sample log–rank statistic. One–sample log–rank tests rather assume that the reference survival curve is a priori known and deterministic (see [2–7, 9]). This ignores that the reference curve resulted from an estimation process, complicates interpretation of the test results and implies an inflation in type I error rate. As lined out in [10], this is a general problem in clinical trials with historical controls.

One aim of this paper is to systematically study the amount of type I error inflation in dependence of the design parameters of the trial. Moreover, we provide a consistent estimate of the factor by which the true variance of the one-sample log–rank statistic is underestimated when reference curve sampling variability is ignored. This allows to construct a random variable $Z$ that explicitly accounts for the sampling variability of the reference curve and thus assures strict type I error rate control.

The paper is organized as follows. After settling notation and the testing problem, we derive a consistent estimate of the actual variance of the one-sample log-rank statistic when the reference cumulative hazard function is estimated non–parametrically from historic data using the Nelson–Aalen estimator. We continue with a simulation study which sheds light on the amount of type I error rate inflation of the one-sample log-rank test when the reference curve sampling variability is neglected in the test statistic. As a tool for planning a one-armed survival study, we then provide a formula that can be used to estimate the inflation based on the historical data and the design parameters of a new study. This instrument is also applied in a case study using a real world data example. We conclude with a discussion of our results and future research. Mathematical proofs are shifted to S1 Appendix.

## General aspects

### Notation

We assume that historic data on standard of care (group A) is available and consider a single arm survival trial where survival data from a new treatment is collected (group B). Let $\mathcal{N}_x$ denote the set of patients from group $x = A, B$, $n_x := |\mathcal{N}_x|$ the number of such patients, and $n := n_A + n_B$ the total number of patients. In particular, we denote by $\pi := n_B/n_A$ the treatment group allocation ratio.

The parameter $n$ will index the arrival process and asymptotic results will be derived in the limit $n \rightarrow \infty$. Accordingly, we assume that the group sizes grow uniformly as total sample size increases, i.e. we assume $\pi$ as a fixed constant.

We denote by $T_{x,i}$ or $C_{x,i}$ the time from entry to event or censoring for patient $i$ from group $x = A, B$, respectively. Let $X_{x,i} := T_{x,i} \wedge C_{x,i}$ denote the minimum of both. As usual, we assume that the $T_{x,i}$ and $C_{x,i}$ are mutually independent (non–informative censoring). Based on the observed data, we calculate the *number of events* from treatment group $x = A, B$ up to study time $s \geq 0$ as

$$N_x(s) := \sum_{i \in \mathcal{N}_x} N_{x,i}(s), \quad N_{x,i}(s) := I(T_{x,i} \leq s, T_{x,i} \leq C_{x,i}), \tag{1}$$

and the *number at risk* $Y_x(s) := \sum_{i \in \mathcal{N}_x} I(T_{x,i} \wedge C_{x,i} \geq s)$ by study time $s \geq 0$ in treatment group $x = A, B$. Let $J_x(s) := I(Y_x(s) > 0)$ indicate whether there are still patients at risk in treatment group $x$ by study time $s$. As usual, we let $\lambda_x(s) := \lim_{\Delta \rightarrow 0} P(s \leq T_{x,i} < s + \Delta | T_{x,i} \geq s)/\Delta$ denote

the hazard of a patient from treatment group $x = A, B$. We denote by $\Lambda_x(s) := \int_0^s \lambda_x(u)du$ the corresponding cumulative hazard function for treatment group $x = A, B$, respectively. Finally, we denote by $f_{T_x}, F_{T_x}, S_{T_x}$ (or $f_{C_x}, F_{C_x}, S_{C_x}$) the density, distribution function and survival function of the time to event $T_{x,i}$ (or time to censoring $C_{x,i}$) in treatment group $x = A, B$. Notice that $\lambda_x, \Lambda_x, f_{T_x}, F_{T_x}, S_{T_x}$ and $f_{C_x}, F_{C_x}, S_{C_x}$ are assumed to coincide for all patients from the same treatment group.

We will also need the Nelson–Aalen estimator (see [11, 12])

$$\hat{\Lambda}_x(s) := \int_0^s \frac{J_x(u)}{Y_x(u)} dN_x(u) \equiv \sum_{i \in \mathcal{N}_x, N_{x,i}(s)=1} \frac{J_x(T_{x,i})}{Y_x(T_{x,i})} \tag{2}$$

of the cumulative hazard function $\Lambda_x(s)$ for group $x = A, B$, and the corresponding estimator of the variance function

$$\hat{\sigma}_x^2(s) := n_x \cdot \int_0^s \frac{J_x(u)}{Y_x^2(u)} dN_x(u) \equiv n_x \cdot \sum_{i \in \mathcal{N}_x, N_{x,i}(s)=1} \frac{J_x(T_{x,i})}{Y_x^2(T_{x,i})}. \tag{3}$$

We consider $N_x, Y_x, J_x, \hat{\Lambda}_x$ and $\hat{\sigma}_x^2$ as stochastic processes in study time $s \geq 0$. Notice that we define $0/0 := 0$ whenever formal division of zero by zero occurs in a mathematical expression. Any stochastic process and martingale in this manuscript is regarded w.r.t. the filtration generated by the observable survival times which is defined at the beginning of Appendix A in S1 Appendix.

## Motivation

The classical one–sample log–rank test (see [1, 2]) assesses the null hypothesis

$$H_{ref} : \Lambda_B(s) = \Lambda_0(s) \quad \text{for all } s \in [0, s_{\max}],$$

that the hazard $\Lambda_B$ of patients from the experimental group B coincides with some prefixed reference hazard $\Lambda_0$ on some prefixed observation horizon $0 \leq s \leq s_{\max}$. Common basis for construction of the one–sample log–rank test is the stochastic process $M_0(s) := n_B^{-1/2}[N_B(s) - \sum_{i \in \mathcal{N}_B} \Lambda_0(s \wedge X_{B,i})]$. When $H_{ref}$ holds true, $M_0$ is (known to be) a mean–zero martingale whose variance $\Sigma_{\text{OSLR}}^2(s) := \int_0^\infty F_{T_B}(s \wedge u)f_{C_B}(u)du$ may consistently be estimated by $n_B^{-1}N_B(s)$ or $n_B^{-1}\sum_{i \in \mathcal{N}_B} \Lambda_0(s \wedge X_{B,i})$ (see e.g. [13]). A standardized version of the one–sample log–rank statistic is then given by $M_0(s_{\max})/(n_B^{-1}N_B(s_{\max}))$ resp. $M_0(s_{\max})/(n_B^{-1}\sum_{i \in \mathcal{N}_B} \Lambda_0(s_{\max} \wedge X_{B,i}))$ which are asymptotically standard normally distributed under the null hypothesis $H_{ref}$.

In clinical practice, the reference curve $\Lambda_0$ is typically intended to represent the survival under standard of care $\Lambda_A$ i.e. it is aimed that $\Lambda_0 \equiv \Lambda_A$. Accordingly, one is actually interested in the two-sided null hypothesis

$$H_0 : \Lambda_B(s) = \Lambda_A(s) \quad \text{for all } s \in [0, s_{\max}]$$

which is the intersection of the two one-sided hypotheses

$$H_{0,\text{sup}} : \Lambda_B(s) \leq \Lambda_A(s) \quad \text{for all } s \in [0, s_{\max}] \text{ and}$$

$$H_{0,\text{inf}} : \Lambda_B(s) \geq \Lambda_A(s) \quad \text{for all } s \in [0, s_{\max}].$$

In this context however, the immediate difficulty is that the true cumulative hazard $\Lambda_A$ under standard of care is unknown, and thus in practice cannot be used as a reference function

in the one–sample log–rank test. To get around this problem it is common practice in the implementation of the classical one–sample log–rank test to estimate $\Lambda_A$ from historic data, and to choose the obtained estimate $\hat{\Lambda}_A$ for $\Lambda_A$ as reference cumulative hazard function, while pretending (i) that $\hat{\Lambda}_A$ is deterministic and (ii) that $\hat{\Lambda}_A$ coincides with $\Lambda_A$. Consequently, the practical implementation of the classical log–rank test often is to consider the processes $\hat{M}_0(s) := n_B^{-1/2}[N_B(s) - \sum_{i \in \mathcal{N}_B} \hat{\Lambda}_A(s \wedge X_{B,i})]$, $\hat{\Sigma}_{\text{OSLR},1}(s) = n_B^{-1} N_B(s)$, $\hat{\Sigma}_{\text{OSLR},2}(s) = n_B^{-1} \sum_{i \in \mathcal{N}_B} \hat{\Lambda}_A(s \wedge X_{B,i})$ and to use the statistic

$$Z_{\text{OSLR},i} \frac{\hat{M}_0(s_{\max})}{\hat{\Sigma}_{\text{OSLR},i}(s_{\max})} \tag{4}$$

for $i = 1$ or $i = 2$ for testing the null hypothesis $H_0$, while additionally pretending that still $Z_{\text{OSLR},i} \sim \mathcal{N}(0, 1)$ under $H_0$. In doing so, note that the maximum observation time in group B must be smaller than the maximum observation duration in the control group so that the above comparison with the estimator $\hat{\Lambda}_A$ from the control group can be made at all. However, this approach ignores that the estimator $\hat{\Lambda}_A$ for $\Lambda_A$ is in fact random and thus contributes additional variance to the test statistic. Consequently, $\hat{\Sigma}_{\text{OSLR},i}(s_{\max})$ underestimates the true variance of $\hat{M}_0(s_{\max})$. Hence, $Z_{\text{OSLR},i}$ in fact fails to be standard normally distributed under $H_0$ and inflation of the type I error rate results. The aim of the following is to systematically study the extent of this type I error rate. In a first step, a correct estimator of the variance of the process $\hat{M}_0$ has to be worked out.

## Revisiting the one–sample log–rank test statistic

Consider the stochastic processes

$$
\begin{aligned}
\hat{M}_0(s) &:= n_B^{-1/2} \left[ N_B(s) - \sum_{i \in \mathcal{N}_B} \hat{\Lambda}_A(s \wedge X_{B,i}) \right] \\
\hat{\Sigma}_1^2(s) &:= n_B^{-1} N_B(s) + n_B^{-1} n_A^{-1} \sum_{i,j \in \mathcal{N}_B} \hat{\sigma}_A^2(s \wedge X_{B,i} \wedge X_{B,j}) \\
\hat{\Sigma}_2^2(s) &:= n_B^{-1} \sum_{i \in \mathcal{N}_B} \hat{\Lambda}_A(s \wedge X_{B,i}) + n_B^{-1} n_A^{-1} \sum_{i,j \in \mathcal{N}_B} \hat{\sigma}_A^2(s \wedge X_{B,i} \wedge X_{B,j})
\end{aligned}
\tag{5}
$$

with $N_B$, $\hat{\Lambda}_A$ and $\hat{\sigma}_A^2$ according to (1), (2) and (3). Assume that the null hypothesis $H_0 : \Lambda_B(s) = \Lambda_A(s)$ for all $0 \leq s \leq s_{\max}$ holds true. Then by Theorem 1 (see S1 Appendix) $\hat{M}_0$ is a mean–zero martingale and for each fixed $s_{\max} \geq s > 0$ we have $\hat{M}_0(s) \xrightarrow{d} \mathcal{N}(0, \Sigma^2(s))$ in distribution as $n \to \infty$, where $\Sigma(s) := \text{plim}_{n \to \infty} \hat{\Sigma}_i(s) = \lim_{n \to \infty} E[\hat{\Sigma}_i(s)]$ for $i \in \{1, 2\}$ (see S1 Appendix, Lemma 1 and Corollary 1). In particular, we conclude that $\hat{\Sigma}_1^2$ and $\hat{\Sigma}_2^2$ are consistent estimators of the variance of $\hat{M}_0$, and that the random variable

$$Z_i := \frac{\hat{M}_0(s_{\max})}{\hat{\Sigma}_i(s_{\max})} \stackrel{H_0}{\sim} \mathcal{N}(0, 1) \tag{6}$$

for $i \in \{1, 2\}$ is approximately standard normally distributed under the null hypothesis $H_0$ if $S_{X_A}(s_{\max}) = S_{T_A}(s_{\max}) S_{C_A}(s_{\max}) =: p_0 > 0$ (see Theorem 1 in S1 Appendix). A sufficient condition for $p_0 > 0$ is as follows: Let $a_B$ and $f_B$ denote the length of accrual and follow–up period in

group B and let $s_{\max} = a_B + f_B$. Let $s_{A,\max}$ denote the maximum observation time in the historic control group A, i.e. $s_{A,\max} = \max_{i \in \mathcal{N}_A} X_{A,i}$. Then $p_0 > 0$ if $s_{\max} < s_{A,\max}$.

Also note that the factor $n_A^{-1}$ in the second summand of $\hat{\Sigma}_i^2(s_{\max})$ cancels out with the factor $n_A$ from the definition of $\hat{\sigma}_A^2$ and the factor $n_B^{-1/2}$ from both the numerator and the denominator of $Z_1$ cancel each other out.

In contrast, the standard one–sample log–rank test statistic at $s_{\max}$ is

$$Z_{\mathrm{OSLR},i} := \frac{\hat{M}_0(s_{\max})}{\hat{\Sigma}_{\mathrm{OSLR},i}(s_{\max})} \tag{7}$$

for an $i \in \{1, 2\}$. The standard one–sample log–rank test of the two-sided null hypothesis $H_0$ is by definition considered to be significant to the level $\alpha$ whenever

$$|Z_{\mathrm{OSLR},i}| \geq \Phi^{-1}\left(1 - \frac{\alpha}{2}\right). \tag{8}$$

Analogously, the one-sided hypotheses $H_{0,\sup}$ or $H_{0,\inf}$ were rejected at the level of $\alpha/2$ by classical one-sample log-rank tests if

$$Z_{\mathrm{OSLR},i} \leq \Phi^{-1}\left(\frac{\alpha}{2}\right) \text{ or } Z_{\mathrm{OSLR},i} \geq \Phi^{-1}\left(1 - \frac{\alpha}{2}\right), \text{ respectively.} \tag{9}$$

It follows directly from the distribution approximation (6), however that $Z_{\mathrm{OSLR},i}$ is in truth not standard normal under the null hypothesis $H_0$, since for both $i \in \{1, 2\}$, $\hat{\Sigma}_{\mathrm{OSLR},i}^2(s_{\max})$ falls short of the consistent variance estiamtors $\hat{\Sigma}_i^2(s_{\max})$ of $\hat{M}_0(s_{\max})$ by the amount $n_B^{-1} n_A^{-1} \sum_{i,j \in \mathcal{N}_B} \hat{\sigma}_A^2(s_{\max} \wedge X_{B,i} \wedge X_{B,j})$ representing the reference curve sampling variability. This results in type I error rate inflation.

The exact amount of the type I error rate inflation is driven by the ratio of the standard deviations

$$R := \Sigma_{\mathrm{OSLR}}(s_{\max})/\Sigma(s_{\max}). \tag{10}$$

This ratio can be consistently estimated by

$$\hat{R}_i := \hat{\Sigma}_{\mathrm{OSLR},i}(s_{\max})/\hat{\Sigma}_i(s_{\max}) \tag{11}$$

for $i \in \{1, 2\}$. The actual type I error rate of the one-sample procedure under $H_0$ can thus be approximated by

$$\alpha_{\mathrm{OSLR}} := 2 \cdot \Phi\left(R \cdot z_{\frac{\alpha}{2}}\right). \tag{12}$$

If recruitment and censoring mechanism were equal in both groups, $R$ would amount to $\sqrt{1/(1+\pi)}$ and the actual type I error level would be inflated to

$$\alpha_{\mathrm{OSLR}} = 2 \cdot \Phi\left(\sqrt{1/(1+\pi)} \cdot z_{\frac{\alpha}{2}}\right). \tag{13}$$

We refer to S1 Appendix for the general case and the derivation of this formula.

In particular the classical one–sample log–rank test procedure (8) exceeds the nominal level $\alpha$ whenever the reference curve sampling variability is large. In this sense the procedure (8) is invalid to test for $H_0$.

In contrast, notice that the two–sample log–rank test would be a valid test for testing the null hypothesis $H_0$ that survival in the new and historic control coincide.

At this point it should be noted that it would be natural to choose the modified test statistic $Z_i$ as a new statistic for testing $H_0$. In a forthcoming paper we will examine its performance regarding type I error rate and power as compared to the two-sample log–rank test. However, these aspects are beyond the focus and scope of this manuscript.

## Simulation study: Effective type I error rate of the one–sample log–rank tsest

### Design

The objective of this simulation study is to quantify the amount of type I error rate inflation, when the reference curve serving as benchmark in the one–sample log–rank test is estimated from historic data, but the reference curve sampling variability is ignored in the test statistic.

In our simulations we focussed on settings of particular practical relevance: Patients were assumed to enter the trial uniformly between year 0 and year $a$ = 2. Accordingly, the calendar times of entry were generated according to a uniform distribution on [0, $a$], i.e. $Y_{x,i} \sim \mathcal{U}(0, a)$. After the end of the accrual period, patients were assumed to be followed up for further $f$ = 3 years, while assuming no loss to follow–up. Hence, we have $C_{x,i} := a + f - Y_{x,i} \sim \mathcal{U}(f, a + f)$ for $x = A, B$. Survival times $T_{A,i}$ in the historic control group $A$ were generated according to a Weibull distribution $\Lambda_A(s) := -\log(S_1) \cdot t^\kappa$ with prefixed shape parameter $\kappa \in \{0.5, 1.0, 2.0\}$ and 1-year survival rate $S_{T_A}(1) = S_1 = 0.5$. Survival times $T_{B,i}$ in the new treatment group $B$ were generated from the same distribution ($\Lambda_B = \Lambda_A$), because our focus is on the type I error rate inflation of the classical one–sample log–rank test when used for testing the null hypothesis $H_0 : \Lambda_B = \Lambda_A$.

To perform the one–sample log–rank test, the group A data was used to calculate the Nelson–Aalen estimate $\hat{\Lambda}_A$ of $\Lambda_A$, and the procedure defined in Eq (8) was applied with a desired two–sided significance level of $\alpha$ = 5% with both variance estimators $\hat{\Sigma}^2_{\text{OSLR},1}$ and $\hat{\Sigma}^2_{\text{OSLR},2}$.

The simulations were used to estimate (i) the empirical type I error rate $\hat{\alpha}$ of the two-sided procedures (8) when used for testing $H_0$ and (ii) the median factors $\hat{R}_i := \hat{\Sigma}_{\text{OSLR},i}(s_{\max})/\hat{\Sigma}_i(s_{\max})$ by which the true standard deviation of the one–sample log–rank statistic $\hat{M}_0$ is underestimated when sampling variability of the reference curve estimate is ignored. Additionally, we study the empirical type I error rates $\hat{\alpha}_{\text{sup}}$ and $\hat{\alpha}_{\text{inf}}$ of the one-sided procedures (9) for testing the two one-sided hypotheses $H_{0,\text{sup}}$ and $H_{0,\text{inf}}$. In order to satisfy the requirements of our asymptotical results, we chose $s_{\max} = a + f - 10^{-8}$.

We used different sample sizes $n_B \in \{25, 50, 100, 200\}$ for group B and allocation ratios $\pi = n_B/n_A \in \{1, 1/2, 1/4, 1/8, 1/16\}$ to study the impact of these parameters on the amount of type I error rate inflation and underestimation of the true variance. Scenarios with $\pi \leq 1/2$ are more likely to reflect common practice as the size of the experimental cohort is typically smaller than the size of the historical control cohort.

For each parameter constellation, we generated 100,000 samples to which we applied the one–sample log–rank test procedures and calculated the underestimation of variance and empirical type I error rates. For this number of samples, the breadth of a 95% confidence interval ranges between 0.0027 and 0.0057 for underlying true rates between 0.05 and 0.3. The results for $\kappa$ = 1 are shown in Tables 1 and 2. The results for $\kappa$ = 0.5 and $\kappa$ = 2 are shifted to Appendix C of S1 Appendix.

**Table 1. Empirical type I error rates under consideration of sampling variability.**

| $n_B$ | $\pi = 1$ | | $\pi = 1/2$ | | $\pi = 1/4$ | | $\pi = 1/8$ | | $\pi = 1/16$ | |
|---|---|---|---|---|---|---|---|---|---|---|
| | $\hat{\alpha}$ | $\hat{R}_i$ | $\hat{\alpha}$ | $\hat{R}_i$ | $\hat{\alpha}$ | $\hat{R}_i$ | $\hat{\alpha}$ | $\hat{R}_i$ | $\hat{\alpha}$ | $\hat{R}_i$ |
| using $\hat{\Sigma}_{\text{OSLR},1}$ as variance estimator | | | | | | | | | | |
| 25 | 0.143 | 0.689 | 0.100 | 0.804 | 0.077 | 0.884 | 0.065 | 0.935 | 0.058 | 0.963 |
| 50 | 0.155 | 0.696 | 0.107 | 0.810 | 0.080 | 0.889 | 0.066 | 0.938 | 0.059 | 0.966 |
| 100 | 0.161 | 0.701 | 0.108 | 0.813 | 0.079 | 0.892 | 0.065 | 0.941 | 0.057 | 0.968 |
| 200 | 0.164 | 0.703 | 0.108 | 0.815 | 0.079 | 0.893 | 0.064 | 0.942 | 0.057 | 0.969 |
| using $\hat{\Sigma}_{\text{OSLR},2}$ as variance estimator | | | | | | | | | | |
| 25 | 0.167 | 0.689 | 0.117 | 0.804 | 0.086 | 0.884 | 0.071 | 0.935 | 0.063 | 0.963 |
| 50 | 0.169 | 0.696 | 0.114 | 0.810 | 0.084 | 0.889 | 0.070 | 0.938 | 0.061 | 0.966 |
| 100 | 0.167 | 0.701 | 0.112 | 0.813 | 0.082 | 0.892 | 0.067 | 0.941 | 0.059 | 0.968 |
| 200 | 0.166 | 0.703 | 0.110 | 0.815 | 0.080 | 0.893 | 0.065 | 0.942 | 0.058 | 0.969 |

(i) Empirical two–sided type I error rates $\alpha$ of test procedure (8) when used for testing $H_0 : \Lambda_B = \Lambda_A$, and (ii) median factors $\hat{R}_i$ as in (11) by which the true standard deviation of the one–sample log–rank statistic $\hat{M}_0$ is underestimated when ignoring the reference curve sampling variability for different parameter constellations of practical relevance. Survival times were Weibull distributed with shape parameter $\kappa = 1$ and 1–year survival rate $S_1 = 0.5$ in the historic control group A and the new treatment group B. Theoretical two–sided significance level: 5%. Underlying sample size of group $B$ is $n_B$ with allocation ratio $\pi = n_B/n_A$ between new and historic groups.

## Results

The classical one–sample log–rank test procedure defined in (8) does not account for sampling variability of the reference curve estimate. This leads to type I error rate inflation when the underlying null hypothesis to be tested is $H_0 : \Lambda_B = \Lambda_A$. As expected, our simulations support that the amount of type I error rate inflation of the one–sample log–rank test is most pronounced when the historic control group is small compared to the new treatment group, i.e. when the allocation ratio $\pi$ is large. For most constellations, the inflation for the test statistics $Z_{\text{OSLR},1}$ slightly decreases with increasing overall sample size $n$ but stabilizes on some level

**Table 2. Empirical one-sided type I error rates under consideration of sampling variability.**

| $n_B$ | $\pi = 1$ | | $\pi = 1/2$ | | $\pi = 1/4$ | | $\pi = 1/8$ | | $\pi = 1/16$ | |
|---|---|---|---|---|---|---|---|---|---|---|
| | $\hat{\alpha}_{\text{inf}}$ | $\hat{\alpha}_{\text{sup}}$ | $\hat{\alpha}_{\text{inf}}$ | $\hat{\alpha}_{\text{sup}}$ | $\hat{\alpha}_{\text{inf}}$ | $\hat{\alpha}_{\text{sup}}$ | $\hat{\alpha}_{\text{inf}}$ | $\hat{\alpha}_{\text{sup}}$ | $\hat{\alpha}_{\text{inf}}$ | $\hat{\alpha}_{\text{sup}}$ |
| using $\hat{\Sigma}_{\text{OSLR},1}$ as variance estimator | | | | | | | | | | |
| 25 | 0.081 | 0.062 | 0.066 | 0.034 | 0.054 | 0.023 | 0.048 | 0.017 | 0.043 | 0.015 |
| 50 | 0.087 | 0.069 | 0.065 | 0.042 | 0.052 | 0.029 | 0.044 | 0.022 | 0.039 | 0.019 |
| 100 | 0.087 | 0.074 | 0.063 | 0.046 | 0.047 | 0.032 | 0.040 | 0.025 | 0.035 | 0.022 |
| 200 | 0.086 | 0.077 | 0.060 | 0.048 | 0.045 | 0.034 | 0.037 | 0.027 | 0.033 | 0.024 |
| using $\hat{\Sigma}_{\text{OSLR},2}$ as variance estimator | | | | | | | | | | |
| 25 | 0.050 | 0.117 | 0.038 | 0.079 | 0.028 | 0.058 | 0.023 | 0.048 | 0.020 | 0.043 |
| 50 | 0.062 | 0.106 | 0.043 | 0.071 | 0.032 | 0.052 | 0.026 | 0.044 | 0.022 | 0.039 |
| 100 | 0.069 | 0.099 | 0.047 | 0.065 | 0.033 | 0.049 | 0.027 | 0.040 | 0.023 | 0.036 |
| 200 | 0.073 | 0.094 | 0.048 | 0.062 | 0.035 | 0.045 | 0.028 | 0.037 | 0.025 | 0.033 |

(i) Empirical one–sided type I error rates $\alpha_1$ and $\alpha_2$ of test procedures (9) when used for testing $H_{0,\text{sup}}$ and $H_{0,\text{inf}}$, respectively, for different parameter constellations of practical relevance. Survival times were Weibull distributed with shape parameter $\kappa = 1$ and 1–year survival rate $S_1 = 0.5$ in the historic control group A and the new treatment group B. Theoretical one–sided significance level: 2.5%. Underlying sample size of group $B$ is $n_B$ with allocation ratio $\pi = n_B/n_A$ between new and historic groups.

above the desired significance level of $\alpha = 5\%$. For the test statistic $Z_{\mathrm{OSLR},2}$ one can observe a slight increase of this inflation with increasing overall sample size and a stabilization on the same level as for $Z_{\mathrm{OSLR},1}$. This supports that the observed type I error rate inflation is primarily not a small sample size phenomenon, but rather due to the underestimation of the variance in the one–sample log–rank statistic. The type I error rate varies furthermore only slightly between the different shape parameters. For ratios $\pi = 1$, the true two-sided type I error rate is approximately three times larger than the desired one (14.3%−16.9% instead of 5% for $\pi = 1$ and $\kappa = 1$). For low allocation ratios as 1/8 or 1/16, the actual two-sided type I error still exceeds the nominal level, but to an extent that might be acceptable for a phase II trial (5.7%−6.3% for $\pi = 1/16$ and $\kappa = 1$; 6.4%−7.1% for $\pi = 1/8$ and $\kappa = 1$). The one-sided type I error rates, however, are quite imbalanced with the direction of imbalance heavily linked to the variance estimator used. This is a well-known phenomenon (see [14]), that affects our simulation results in addition to the neglected variance. Estimation of the variance with the counting process estimator $\Sigma_{\mathrm{OSLR},1}$ leads in the finite sample case to a left-skewed distribution of $Z_{\mathrm{OSLR},1}$ and thus more decisions in favour of the new treatment are made. Estimation with the compensator process via $\Sigma_{\mathrm{OSLR},2}$ in contrast leads to a right-skewed distribution of $Z_{\mathrm{OSLR},2}$. Even for small allocation ratios at $\pi = 1/8$ both tests have an one-sided error rate above 3.7% instead of 2.5% in their corresponding favoured direction. For small historic control groups ($\pi \geq 1/2$) the effect of ignoring reference curve sampling variability on type I error rate inflation predominates these effects of skewness.

Varying the shape parameter $\kappa$ does only change the inflation slightly (see Appendix C in S1 Appendix). This is to be expected as the log-rank test is a rank-based test. By transformations of the time scale, the survival distributions of the different scenarios can be transformed into each other such that only the distributions of entry and censoring times differ between the scenarios. This is reconfirmed by the fact that in case of equal entry and censoring distributions of groups $A$ and $B$ the asymptotical inflation in Eq (13) does only depend on $\pi$ and no other design parameters.

With a view to application of the classical one-sample log-rank test (8) for testing $H_0$ in historically controlled phase II survival trials, our results support that as a rule of thumb choice of the reference curve should be based on a historic control that is at least about 12 times larger than the new experimental trial cohort. According to (13), a factor of at least 12 corresponds to an inflation of the type I error rate to a maximum of 6%. For a stricter type I error rate control one could implement a hybrid testing procedure defined by rejecting $H_0$ when either $Z_{\mathrm{OSLR},1} \geq \Phi^{-1}(1-\alpha/2)$ or $Z_{\mathrm{OSLR},2} \leq \Phi^{-1}(\alpha/2)$. This hybrid testing strategy exploits the skewness of the statistics $Z_{\mathrm{OSLR},i}$ to compensate in parts for the type I error rate inflation due to neglect of the reference curve sampling variability. In our simulations, this strategy yields valid tests of $H_0$ for allocation ratios $\pi \leq 1/8$. If the historic control group A is small ($\pi \geq 1/4$), the null hypothesis of no difference between group $B$ and $A$ should rather be tested by a two–sample log–rank test.

Furthermore, the maximum observation time of the new trial should also be set smaller than the one of the historic control to avoid utilizing the volatile tails of the Kaplan-Meier curve within the test statistic. This is also supported by the calculation of $\alpha_{\mathrm{OSLR}}$ as defined in (12) via (10). The results of this calculation are displayed in Fig 1. The inflated type I error level is plotted as a function of the allocation ratio $\pi$ for three different durations of the follow-up period. As expected, longer observation periods lead to a higher inflation of the type I error rate. This is due to the fact that the estimation of the survival time in group A becomes more volatile at the tail of the distribution which is more frequently utilized in the test statistic for group B if the follow-up duration is extended.

In summary, the simulations support that neglecting the reference curve sampling variability in the classical one–sample log–rank test relevantly compromises type I error rate control

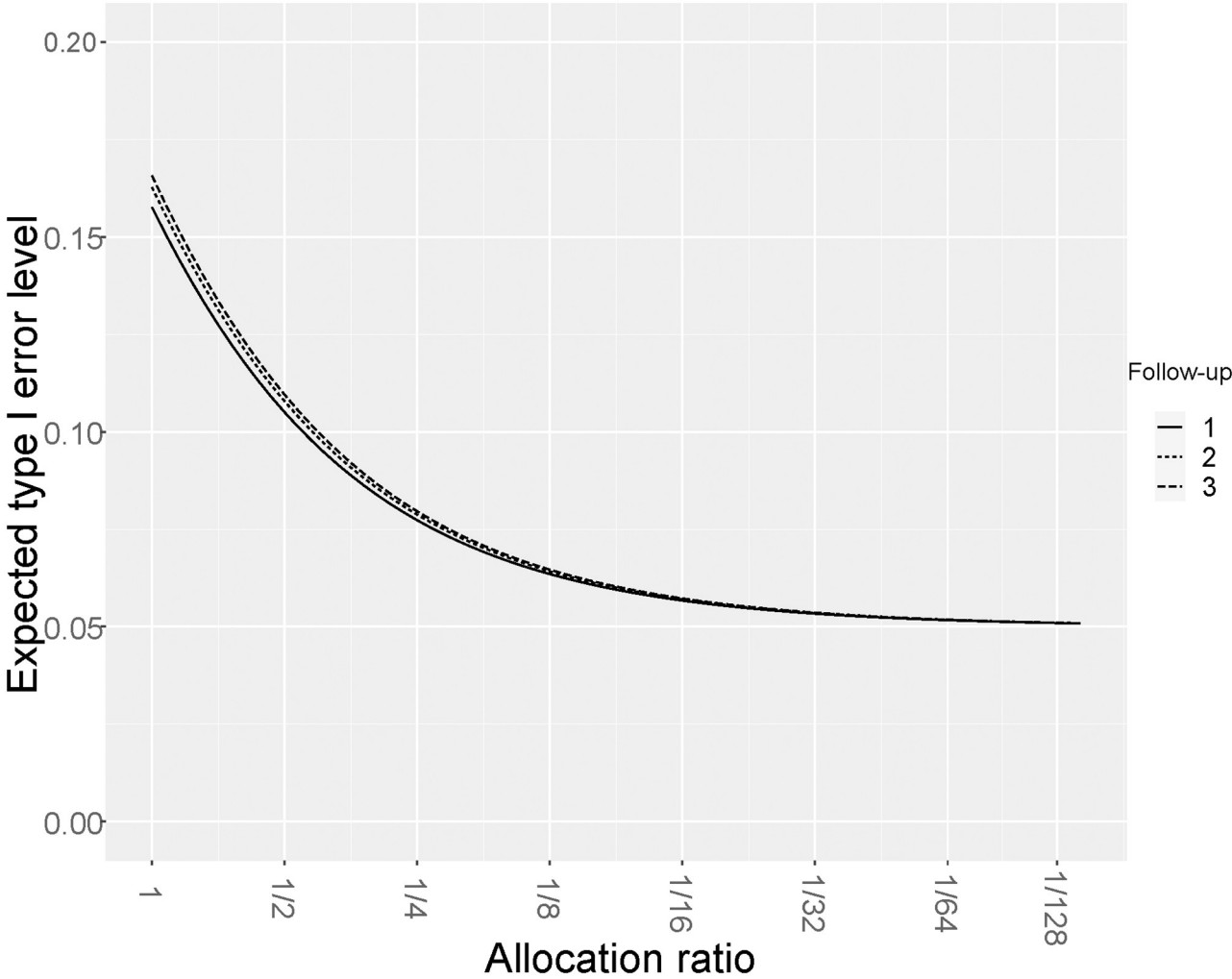

**Fig 1. Type I error rate approximation.** Type I error rate approximation given by $2 \cdot \Phi\left(\Sigma_{\mathrm{OSLR}}(s_{\max})/\Sigma(s_{\max}) \cdot z_{\frac{\alpha}{2}}\right)$ as a function of the allocation ratio $\pi$ for different durations $f_B \in \{1, 2, 3\}$ of the follow-up period in the new trial. Calculations were done for exponentially distributed survival times with a 1 year survival rate of 50%. Accrual $a$ for the historic control and new treatment groups was set to 5 years, follow-up $f_A$ of the historic trial was set to 3 years. To satisfy the conditions of Theorem 1 (see S1 Appendix), we choose $s_{\max} = a + f_B - 10^{-8}$.

when testing null hypotheses $H_0 : \Lambda_B = \Lambda_A$. Notice that the classical one–sample log–rank test only realizes strict type I error rate control for testing the null hypothesis $\tilde{H}_0 : \Lambda_B = \hat{\Lambda}_A$ which, however, detracts from the null hypothesis $H_0 : \Lambda_B = \Lambda_A$ of true interest when random deviation of $\hat{\Lambda}_A$ from $\Lambda_A$ is large.

## A priori estimation of the expected type I error rate inflation

As seen in the preceding simulations, the actual type I error rate of the classical one-sample log-rank test always exceeds the nominal type I error level if the sampling variability of the reference curve is not taken into account. However, the magnitude of this excess depends on the data from the reference cohort as well as the sample size in the new, experimental cohort. In this section, we describe how to estimate the expected amount of type I error rate inflation already at the planning stage of a historically controlled, single-arm survival trial. This allows

an a priori assessment of whether the one-sample log-rank test can be considered appropriate to test $H_0$ in the particular trial setting or whether the use of alternative methods such as the two-sample log-rank test is preferable.

The difference between the test statistic of the classical one-sample log-rank statistic $Z_{\mathrm{OSLR},i}$ from (4) and the asymptotically standard normally distributed random variable $Z_i$ from (6) is the standardization factor in the respective denominators. Let $\hat{R}_i$ from (11) denote the ratio of the standardisation factors without and with consideration of the sampling variability. With the factors $\hat{R}_i$ it is possible to explicitly quantify the expected amount of type I error rate due to neglect of reference curve sampling variability: The actual type I error rate of a two-sided classical one-sample log-rank test with nominal level $\alpha$ is in expectation $E\left[2 \cdot \Phi\left(\hat{R}_i \cdot z_{\frac{\alpha}{2}}\right)\right]$ instead of $\alpha$ when reference curve sampling variability is neglected. The former can be approximated by $2 \cdot \Phi\left(E[\hat{R}_i] \cdot z_{\frac{\alpha}{2}}\right)$. Analogously, $E[\hat{R}_i]$ can be approximated via a first-order Taylor expansion by $\sqrt{E[\hat{\Sigma}_{\mathrm{OSLR},i}^2(s_{\max})]/E[\hat{\Sigma}_i^2(s_{\max})]}$. In the planning stage of a new trial, the historical data (summarized by the set of random variables $\mathcal{D}_A$) is already known and can be taken into account when considering the type I error rate inflation. Conditioning on this we can compute

$$R_{\mathrm{pre}} := \sqrt{\frac{E[\hat{\Sigma}_{\mathrm{OSLR},i}^2(s_{\max})]}{E[\hat{\Sigma}_i^2(s_{\max})|\mathcal{D}_A]}} \tag{14}$$

One should note that according to our calculations in S1 Appendix the asymptotics for both $i \in \{1, 2\}$ lead to the same result. Hence, $R_{\mathrm{pre}}$ is well-defined. The expression given here can immediately be estimated from given historical control data and design parameters of a trial (see Appendix B in S1 Appendix for details). Analogously to (12), the actual type I error rate to be expected is given by

$$\alpha_{\mathrm{pre}} := 2 \cdot \Phi\left(R_{\mathrm{pre}} \cdot z_{\frac{\alpha}{2}}\right). \tag{15}$$

The computations in [6] and the asymptotics of the Nelson-Aalen estimator yield

$$E[\hat{\Sigma}_{\mathrm{OSLR},i}^2(s_{\max})] \approx \int_0^\infty F_{T_B}(s_{\max} \wedge u) dF_{C_B}(u). \tag{16}$$

After another approximation and some computations (see Appendix B in S1 Appendix), we also get

$$
\begin{aligned}
E[\hat{\Sigma}_i^2(s_{\max})|\mathcal{D}_A] \approx \quad & \int_0^\infty F_{T_B}(s_{\max} \wedge u) dF_{C_B}(u) \\
& + 2\pi \cdot \left( \int_0^\infty \hat{\sigma}_A^2(s_{\max} \wedge u) S_{T_B}^2(u) S_{C_B}(u) dF_{C_B}(u) \right. \\
& \left. + \int_0^\infty \hat{\sigma}_A^2(s_{\max} \wedge u) S_{T_B}(u) S_{C_B}^2(u) dF_{T_B}(u) \right)
\end{aligned}
$$

Under the null hypothesis $H_0$, the right hand side can be estimated by plugging in Kaplan-Meier estimates gained from the historic control group A for $F_{T_B}$ respectively $S_{T_B}$. For a given historical control group, these formulas can now be used to compute the type I error inflation due to ignoring reference curve sampling variability. Of course, the treatment group allocation

**Table 3. Apriori estimated type I error rates under consideration of sampling variability.**

| $n_B$ | $\pi = 1$ | | $\pi = 1/2$ | | $\pi = 1/4$ | | $\pi = 1/8$ | | $\pi = 1/16$ | |
|---|---|---|---|---|---|---|---|---|---|---|
| | $\alpha_{\text{pre}}$ | $R_{\text{pre}}$ | $\alpha_{\text{pre}}$ | $R_{\text{pre}}$ | $\alpha_{\text{pre}}$ | $R_{\text{pre}}$ | $\alpha_{\text{pre}}$ | $R_{\text{pre}}$ | $\alpha_{\text{pre}}$ | $R_{\text{pre}}$ |
| 25 | 0.156 | 0.724 | 0.107 | 0.823 | 0.079 | 0.896 | 0.064 | 0.943 | 0.057 | 0.970 |
| 50 | 0.161 | 0.715 | 0.108 | 0.820 | 0.079 | 0.895 | 0.065 | 0.943 | 0.057 | 0.970 |
| 100 | 0.163 | 0.711 | 0.109 | 0.818 | 0.079 | 0.895 | 0.065 | 0.943 | 0.057 | 0.970 |
| 200 | 0.165 | 0.709 | 0.109 | 0.817 | 0.080 | 0.895 | 0.065 | 0.943 | 0.057 | 0.970 |

(i) Median a priori estimates of type I error rate $\alpha_{\text{pre}}$ (see Eq (15)) of test procedure (8) when used for testing $H_0 : \Lambda_B = \Lambda_A$, and (ii) median a priori estimates of underestimation of the standard deviation $R_{\text{pre}}$ (see Eq (14)) of the one–sample log–rank statistic $\hat{M}_0$ when ignoring the reference curve sampling variability for different parameter constellations of practical relevance. Survival times were Weibull distributed with shape parameter $\kappa = 1$ and 1–year survival rate $S_1 = 0.5$ in the historic control group A. Underlying sample size of $n_A = n_B/\pi$ with allocation ratio $\pi$.

ratio $\pi$ is essential for the extent of this inflation. We also applied this a priori estimation in our simulation from the previous section. The results can be found in Table 3. They suggest that the underestimation of variance can be robustly examined based on the historic data before the new group is recruited. A much simpler estimate is provided by formula (13). This is particularly useful when no assumption can be made about recruitment and censoring mechanism in group B. From Fig 3, however, it can be seen that these have a large influence on the actual extent of the type I error rate inflation.

We will now illustrate the influence of basic design parameters on the type I error inflation using a practical example. We employ the setting of the Mayo Clinical trial in primary biliary cirrhosis of the liver (PBC), which is a rare but fatal chronic disease whose cause is still unknown (see [15]). In this double-blinded randomized trial the drug D-penicillamine (DPCA) was compared with a placebo. The study data is publicly available via the survival package in R (see [16, 17]).

Among the 158 patients of the cohort treated with DPCA, 65 died during the trial. The Kaplan-Meier survival curve of these patients can be found in Fig 2. The time scale is given in years. In the same figure, we also display the empirical distribution of the censoring variable $C$ in this cohort. As we will see below, the censoring distribution also plays a crucial role for our computations. We now suppose, that a new treatment becomes available and the data from this new trial shall be used to compare the survival under a new treatment to the survival under historic treatment with DPCA. This shall be accomplished in a trial in which patients are recruited uniformly over a accrual period of length $a$ and then followed-up in an subsequent observation phase of length $f$. The allocation ratio (new to historic cohort) will again be denoted by $\pi$. If one cannot find a suitable parametric model to be fitted to the data, the Kaplan-Meier and Nelson-Aalen estimates (see Fig 2) are employed as reference curves for the one-sample log-rank test, respectively.

Similar to our simulation study, we first investigate the influence of the allocation ratio on the type I error inflation. We choose $\pi \in \{0.01, 0.02, 0.03, \ldots, 1\}$, $a = 2$ and $f \in \{2, 4, 6, 8\}$. Hence, we obtain analysis dates $s_{\text{max}} \in \{4, 6, 8, 10\}$. As the observation period of many patients in the historical reference group exceeds 10 years, we do comply with the requirements of Theorem 1 (see Appendix A of S1 Appendix) here. The results in terms of the actual type I error level of the one-sample log-rank test can be found on the left hand side of Fig 3. For any fixed $f$, the actual type I error level increases nearly linearly with the allocation ratio. the amount of increase additionally depends on the length of the follow-up, where a longer duration of the follow-up period leads to steeper increases.

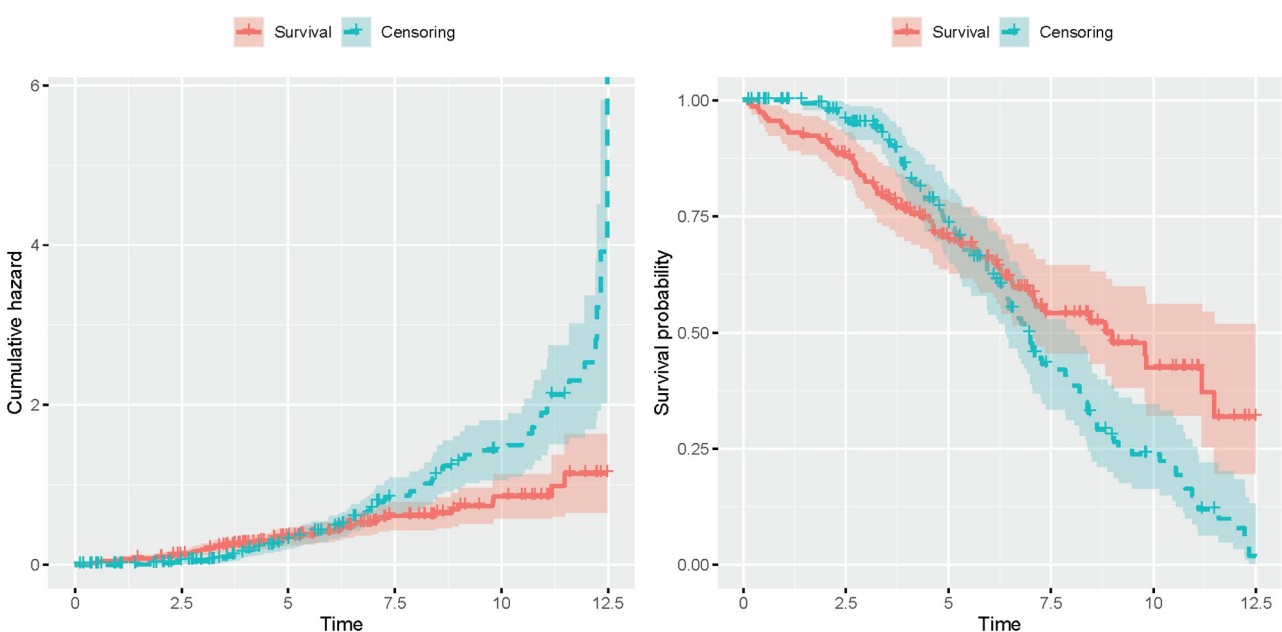

**Fig 2. Distribution of survival and censoring variable.** Distribution of overall survival and censoring in the cohort treated with DPCA of the Mayo Clinical trial in primary biliary cirrhosis. Left: Cumulative hazards according to the Nelson-Aalen estimator. Right: Survival distributions according to the Kaplan-Meier estimator

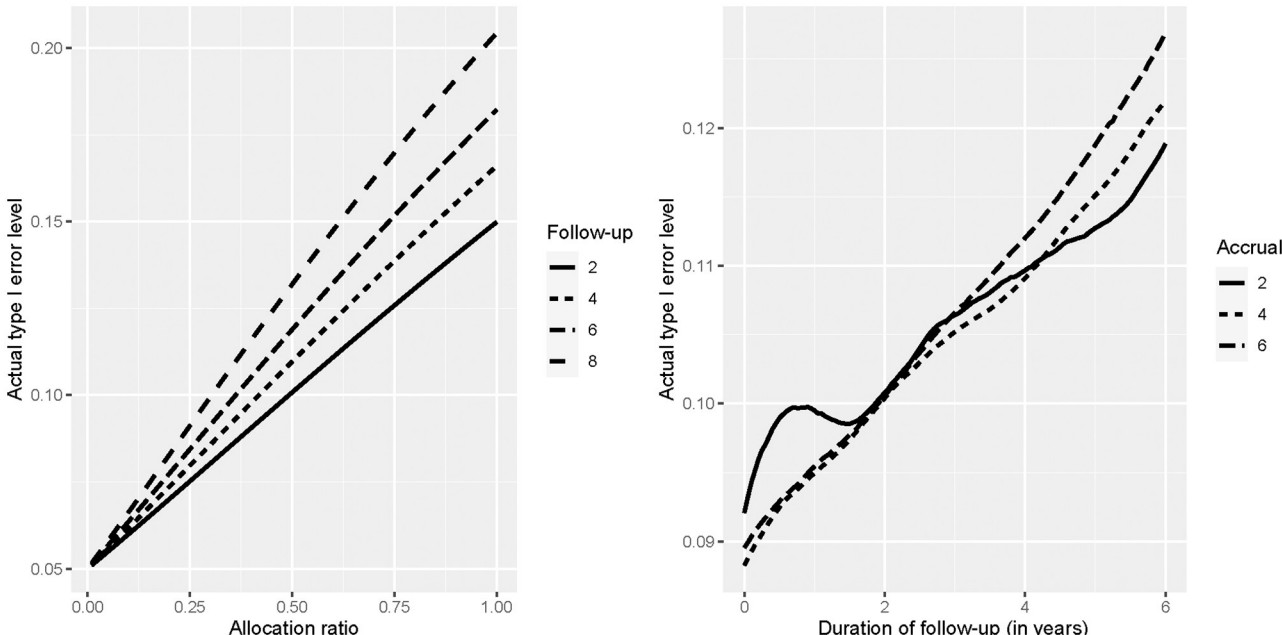

**Fig 3. Type I error inflation.** Actual type I error levels of the classical one-sample log-rank test when sampling variability of the reference curve is ignored. Left: Variation of the allocation ratio with fixed accrual duration $a$ and four different durations of the follow-up period $f$. Right: Variation of the length of the follow-up period $f$ for a fixed allocation ratio $\pi$ and three different durations of the accrual period $a$.

We take a closer look at the role of the trial overall trial duration $a + f$ next. As already seen in the first part, longer trials lead to a larger inflation of the type I error levels. To analyse this dependence, we now choose $\pi = 0.5$, $a \in \{2, 4, 6\}$ and $f \in \{0, 0.05, 0.1, \ldots, 6\}$. The results can be found on the right hand side of Fig 3. As we can see, trials with a longer overall duration $a + f$ lead to larger type I error inflation. This effect is most substantial if the overall duration of the new trial is close to the longest observation in the historic data set (in our example about 12.5 years). The reason is that in this case, the testing procedure needs to utilize parts of the tail of the Nelson-Aalen estimator which are based on a small proportion of patients and thus are affected by a high amount of variability. This stresses the importance of the frame condition that the available follow–up data for patients from the historic group should be substantially longer than the desired length of the new trial, if the reference survival curve is estimated from historic data. However, the inflation of the type I error rate neither behaves completely monotonically in the accrual duration $a$ nor in the follow-up duration $f$. Even if the variance estimators $\hat{\Sigma}_{\mathrm{OSLR},i}^2$ of the one–sample log–rank test and the additional variance $\hat{\Sigma}_i^2 - \hat{\Sigma}_{\mathrm{OSLR},i}^2$ from consideration of the reference curve sampling variability increase monotonically in $a$ and $f$, their ratio can increase if the increase of the former is steeper than the increase of the latter. Nevertheless, there is a clear tendency towards a larger inflation of the type I error rate if either $a$ or $f$ increases.

## Discussion

Traditional one–sample log–rank tests compare the survival function of an experimental treatment to a prefixed reference survival curve, which typically represents the expected survival under standard of care. Choice of the reference survival curve is commonly based on historic data on standard therapy and thus prone to sampling error. Nevertheless, traditional one–sample log–rank tests do not account for this variance of the reference curve estimator, but rather assume that the reference curve is deterministic.

Ignoring the sampling variability however, leads to an inflation of the type I error rate. The extent of this inflation depends in particular on the relative size of the historic control cohort compared to the new treatment cohort. A major objective of this paper was to work out recommendations on the size of the historic control group such that the type I error inflation remains within an acceptable range. In this regard, our simulations support that the classical one-sample log-rank test is adequate for two-sided type I error rate control if the historical control cohort is large enough. If the desired significance level is 5%, Eq (12) suggests that this historic control cohort should be at least 12 times larger than the new cohort ($\pi \leq 1/12$) to assure that the type I error rate is not inflated beyond 6%. Additionally, the available follow–up data for patients from the historic group should be substantially longer than the desired length of the new trial (see Fig 1 and Results). For stricter type I error rate control one could use a hybrid strategy defined by rejecting $H_0$ whenever $Z_{\mathrm{OSLR},1} \geq \Phi^{-1}(1-\alpha/2)$ or $Z_{\mathrm{OSLR},2} \leq \Phi^{-1}(\alpha/2)$. This strategy exploits the skewness of the distribution of different versions of the one-sample log-rank test statistic in order to compensate in parts the type I error rate inflation due to neglect of reference curve sampling variability. In our simulations, this hybrid strategy yields satisfactory type I error rate performance for allocation ratios $\pi \leq 1/8$.

In this respect, it seems advisable to use the classical two-sample log-rank test (see [18]) if these conditions are not met and the proportional hazards assumption can be made. There, the variability in the data of the reference group is naturally taken into account. However, one must be careful here as well, since compliance with the type I error rate is not given in case of small sample sizes or unbalanced groups [19] as in some scenarios of our simulations. However, such issues can be solved by the application of resampling-based tests [20].

We also provided a consistent estimate of the actual variance of the one–sample log–rank statistic when reference curve sampling variability is taken into account. This allows to construct a random variable $Z_i$ (see Eq (6)) that is asymptotically standard normally distributed under the null hypothesis $H_0 : \Lambda_B = \Lambda_A$. $Z_i$ thus yields a test of $H_0$ that may be viewed as an alternative to the two-sample log–rank test for $H_0$. Planning and performance of this new test as compared to the two–sample log–rank test will be contents of a forthcoming paper.

Conceptually, this construction of our random variable $Z_i$ also sheds light on a general strategy for lifting existing methodology for single–arm survival trials to a randomized, multi–arm setting. This might be of interest for designing confirmatory survival trials with interim analyses. Performance of interim analyses in clinical trials is of ethical and economic interest. On the one hand, interim analyses enable faster decisions regarding rejection or acceptance of the underlying null hypothesis when the treatment effect is larger or smaller than initially expected. Moreover, interim analyses offer the possibility for data dependent modifications of the trial (e.g. sample size recalculation) in the case of new insights, thus increasing the prospects of the trial. Trial designs with interim analyses offering such kind of flexibility at full type I error rate control are commonly referred to as *confirmatory adaptive designs* [21, 22]. Advanced one-sample methodology as in [23] might be transformed to be applicable in multi-arm settings in this way to address still existing problems when it comes to the use of additional information in interim analyses (see [24]).

Similarly, weighted one-sample log-rank tests as in [25] which are better suited for the detection of late or early effects can also be analyzed with the methods proposed here. Corresponding weights can be introduced to $Z_{\text{OSLR},i}$ (see (4)) resp. $Z_i$ (see (6)) for $i \in \{1, 2\}$ by multiplying them with the event indicator functions, inserting them into the counting process integral (2) of the Nelson-Aalen estimator and inserting its square into the counting process integral (3) of the variance estimator.

Going beyond our research, we have to point out that we did not consider the problem of confounding in historically controlled trials here. This occurs if the characteristics of the historical control cohort and the cohort of the new study differ substantially. Extreme caution is therefore required when selecting a historical control. In [26, 27], several criteria under which a historical control cohort appears suitable, are given. Of course, known confounders can also be taken into account by choosing an adequate analysis technique. This can be achieved by stratification of the two cohorts or, if appropriate, a Cox proportional hazards model. However, this will be content of future research. The objective of this paper is to provide methodology for accounting for sampling variability of the reference curve in classical one-sample log-rank tests, and illustrate the drastic consequences of neglect of reference curve sampling variability on type I error rate control.

## Supporting information

**S1 File. R code.** Supplementary R code for the estimation of type I error rate inflation via a Monte Carlo simulation.
(ZIP)

**S2 File. R code.** Supplementary R code for a priori estimation of type I error rate inflation given the data from a historic control group.
(R)

**S1 Appendix. Mathematical details.** Mathematical statements and corresponding proofs.
(PDF)

## Acknowledgments

We thank three anonymous reviewers and the editor for their helpful comments, which helped improve the content and presentation of the manuscript. We acknowledge support from the Open Access Publication Fund of the University of Muenster.

## Author Contributions

**Conceptualization:** Jannik Feld.

**Formal analysis:** Moritz Fabian Danzer, Jannik Feld.

**Funding acquisition:** Rene Schmidt.

**Investigation:** Moritz Fabian Danzer, Jannik Feld.

**Methodology:** Moritz Fabian Danzer, Jannik Feld.

**Project administration:** Moritz Fabian Danzer, Jannik Feld.

**Resources:** Moritz Fabian Danzer, Jannik Feld.

**Software:** Moritz Fabian Danzer, Jannik Feld, Rene Schmidt.

**Supervision:** Andreas Faldum, Rene Schmidt.

**Validation:** Rene Schmidt.

**Visualization:** Moritz Fabian Danzer, Jannik Feld.

**Writing – original draft:** Moritz Fabian Danzer, Jannik Feld, Rene Schmidt.

**Writing – review & editing:** Moritz Fabian Danzer, Jannik Feld, Andreas Faldum, Rene Schmidt.

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
