## [Decision Letter · Decision Letter 0]

21 Apr 2022

PONE-D-22-01927Reference Curve Sampling Variability in One-Sample Log-Rank TestsPLOS ONE

Dear Dr. Jannik,

Thank you for submitting your manuscript to PLOS ONE. After careful consideration, we feel that it has merit but does not fully meet PLOS ONE’s publication criteria as it currently stands. Therefore, we invite you to submit a revised version of the manuscript that addresses the points raised during the review process. The scope, the structure and the main content of this new manuscript are now adequate. However, there are some major (time window, improved discussion of Table 1, appropriate type I error rates in both tails) and a number of minor issues which should be addressed (see comments of all 3 reviewers).

We look forward to receiving your revised manuscript.

Best wishes,

Ralf

Ralf Bender, Ph.D.

Academic Editor

PLOS ONE

Journal Requirements:

2. Please note that PLOS ONE has specific guidelines on code sharing for submissions in which author-generated code underpins the findings in the manuscript. In these cases, all author-generated code must be made available without restrictions upon publication of the work. Please review our guidelines at https://journals.plos.org/plosone/s/materials-and-software-sharing#loc-sharing-code and ensure that your code is shared in a way that follows best practice and facilitates reproducibility and reuse

"The work of Moritz Fabian Danzer was funded by the German Science Foundation

(Deutsche Forschungsgemeinschaft, DFG, grant number 413730122)"

"The work of MFD was funded by the German Science Foundation (Deutsche Forschungsgemeinschaft, DFG, https://www.dfg.de, grant number 413730122). The funders had no role in study design, data collection and analysis, decision to publish, or preparation of the manuscript."

Reviewers' comments:

Reviewer's Responses to Questions

**Comments to the Author**

1. Is the manuscript technically sound, and do the data support the conclusions?

Reviewer #1: Yes

Reviewer #2: Yes

Reviewer #3: Yes

2. Has the statistical analysis been performed appropriately and rigorously? 

Reviewer #1: Yes

Reviewer #2: Yes

Reviewer #3: Yes

3. Have the authors made all data underlying the findings in their manuscript fully available?

Reviewer #1: Yes

Reviewer #2: Yes

Reviewer #3: Yes

4. Is the manuscript presented in an intelligible fashion and written in standard English?

Reviewer #1: Yes

Reviewer #2: Yes

Reviewer #3: Yes

5. Review Comments to the Author

Reviewer #1: As suggested in the former round, the authors focused now more on the one-sample setting and explain the effect

of the uncertainty when the reference group is "estimated" from historic data. The comparison of their new

adjusted statistic to a classical two-sample setting is referred to a future project. Thus, the second major point of

my last review is postponed to the future. However, I have still some concerns related to my first major point, which I guess can rather easily be solved or adressed.

1) The Time-window for the analysis

(i) The authors decided not the present the actual conditions on s_max, the upper limit of the analysis window, but prefer a verbal description. "In particular, we require the time

window of observation in the interventional group to be smaller than that of the control group". Maybe the authors can convince me from the opposite but this is not the same as

the condition in Theorem 1 and 2 in the appendix. There it is said that (a) S_{XA}(s0) = S_{TA}(s0)S_{CA}(s0) is required. Here, the censoring also plays an important role. Does the

authors suppose that the censoring in both groups are the same? Otherwise, I do not see how the upper verbal conditions implies the formal condition (a).

(ii) In their response to my former comment, the authors state

"Nevertheless, we have kept the formal argument ”∞” in formulas (5), (7) and (8) to show that no data

from any group is discarded in the final analysis if the condition addressed is complied with."

From a theoretical point of view, the time window must be specified in advance before the data is collected. Since s_max need to be chosen such

that P(X > s_max)>0, it is expected that there are also observations larger than s_max. These observations are not completely excluded because they

are needed in the calculation for the Nelson-Aalen estimator. I understand that the restriction to s_max is not so nice, especially in the

formulas, but it is what the theory gives you. Thus, I would prefer that it is mentioned appropriately in the paper.

The restriction to s_max is also important for the so-called restricted mean survival time (RMST, see e.g. several papers of Royston). So it is not completely

new and also accepted (at least from my point of view). There are also some discussion how to avoid the restrictions, see Tian et al (2020) for the

RMST or Wang (1987) & Stute and Wang (1993) for uniform consistency of the Kaplan–Meier, but in both cases conditions on the censoring distribution are needed.

(iii) How is s_max chosen in the data example and for the simulations? The conditions on s_max ensure that the variance does not "explode", right? Is there a guarentee that this does not happen in the simulations? When this is the case but no s_max is chosen, this would also be fine for me when the authors appropriately explain it in the respective section.

2) Minor Remark/Question:

The log-rank test is known to be optimal for proportional hazard alternatives. But when early or late differences are expected including an appropriate weight in the statistic can lead to a siginifcant benefit in terms of power. I wonder whether such weights can also be implemented in the new proposal. When there is a quick solution, the authors may add a small discussion to the paper, otherwise please ignore the comment.

References

Tian L, Jin H, Uno H, et al. On the empirical choice of the time window for restricted mean survival time. Biometrics 2020.

Wang, J.-G. (1987). A note on the uniform consistency of the Kaplan–Meier estimator. The Annals of Statistics, 15(3), 1313–1316.

Stute, W., Wang, J.-L. (1993). The strong law under random censorship. The Annals of Statistics, 21(3), 1591–1607.

Reviewer #2: see the attached file

Reviewer #3: See attached report.

Here is some content of the attached report to meet the required character count:

The authors explain how a one-sample log-rank test may be used to test whether the survival distribution for a new sample is the same as a reference survival distribution based on historical data. They point out that the standard method of testing does not adjust for error in estimating the reference distribution and, thus, the probability of finding a difference when both the new sample and the historical data follow the same distribution is liable to be inflated above the nominal type I error rate. The main contribution of the paper is to quantify the possible type I error rate inflation.

The description of methods is rather technical.

I believe the main contribution of the paper is the set of results presented in Table 1. Thus, a clear discussion of the patterns in this table and their generalisability to other cases is paramount.

6. PLOS authors have the option to publish the peer review history of their article (what does this mean?). If published, this will include your full peer review and any attached files.

Reviewer #1: No

Reviewer #2: No

Reviewer #3: No

---

## [Author Response · Author response to Decision Letter 0]

29 May 2022

We attached a pdf document with detailed answers to all reviewer comments.

---

## [Decision Letter · Decision Letter 1]

24 Jun 2022

Reference Curve Sampling Variability in One-Sample Log-Rank Tests

PONE-D-22-01927R1

Dear Moritz,

We’re pleased to inform you that your manuscript has been judged scientifically suitable for publication and will be formally accepted for publication once it meets all outstanding technical requirements.

Best wishes,

Ralf

Ralf Bender, Ph.D.

Academic Editor

PLOS ONE

Additional Editor Comments (optional):

1) Please add the missing argument in the formula on page 5, line 122.

2) The formatting of the references is not consistent (upper-lower case in article titles, journal abbreviations).

Reviewers' comments:

Reviewer's Responses to Questions

**Comments to the Author**

1. If the authors have adequately addressed your comments raised in a previous round of review and you feel that this manuscript is now acceptable for publication, you may indicate that here to bypass the “Comments to the Author” section, enter your conflict of interest statement in the “Confidential to Editor” section, and submit your "Accept" recommendation.

Reviewer #1: All comments have been addressed

Reviewer #2: All comments have been addressed

2. Is the manuscript technically sound, and do the data support the conclusions?

Reviewer #1: Yes

Reviewer #2: Yes

3. Has the statistical analysis been performed appropriately and rigorously? 

Reviewer #1: Yes

Reviewer #2: Yes

4. Have the authors made all data underlying the findings in their manuscript fully available?

Reviewer #1: Yes

Reviewer #2: Yes

5. Is the manuscript presented in an intelligible fashion and written in standard English?

Reviewer #1: Yes

Reviewer #2: Yes

6. Review Comments to the Author

Reviewer #1: I thank the authors for the two revision rounds. All my points are now apropriately addressed. All the best also for

the other project regarding two-sample testing, here I think the option of a different weight is also of interest! Anyway,

for me, the paper is ready to go.

A typo:

line 122: the argument of the max_{i in N_A}(?) is missing.

Reviewer #2: Thank you for the careful revision.

Apart from only a minor comment, I have nothing else to add.

On p. 5, line 123, the "maximand" seems to be missing.

7. PLOS authors have the option to publish the peer review history of their article (what does this mean?). If published, this will include your full peer review and any attached files.

Reviewer #1: No

Reviewer #2: No

---

## [Editor Report · Acceptance letter]

13 Jul 2022

PONE-D-22-01927R1 

Reference curve sampling variability in one–sample log–rank tests 

Dear Dr. Danzer:

I'm pleased to inform you that your manuscript has been deemed suitable for publication in PLOS ONE. Congratulations! Your manuscript is now with our production department. 

Kind regards, 

on behalf of

Professor Ralf Bender 

Academic Editor

PLOS ONE